# Two sets of bias-corrected regional UK Climate Projections 2018 (UKCP18) of temperature, precipitation and potential evapotranspiration for Great Britain

Nele Reyniers[1, 2], Qianyu Zha[1, 3], Nans Addor[4, 5], Timothy J Osborn[1, 2, 6], Nicole Forstenhäusler[1, 3], and Yi He[1, 3]

[1]School of Environmental Sciences, University of East Anglia, Norwich, UK
[2]Climatic Research Unit, School of Environmental Sciences, University of East Anglia, Norwich, UK
[3]Tyndall Centre for Climate Change Research, School of Environmental Sciences, University of East Anglia, Norwich, UK
[4]Fathom, Bristol, UK
[5]University of Exeter, Exeter, UK
[6]Water security research centre, University of East Anglia, Norwich, UK

**Correspondence:** Qianyu Zha (q.zha@uea.ac.uk)

**Abstract.** The United Kingdom Climate Projections 2018 (UKCP18) regional climate model (RCM) 12km regional perturbed physics ensemble (UKCP18-RCM-PPE) is one of the three strands of the latest set of UK national climate projections produced by the UK Met Office. It has been widely adopted in climate impact assessment. In this study, we report biases in the raw UKCP18-RCM simulations that are significant and are likely to deteriorate impact assessments if they

are not adjusted. Two methods were used to bias-correct UKCP18-RCM: non-parametric quantile mapping using empirical quantiles and a variant developed for the third phase of the Inter-Sectoral Impact Model Intercomparison Project (ISIMIP) designed to preserve the climate change signal. Specifically, daily temperature and precipitation simulations for 1981 to 2080 were adjusted for the 12 ensemble members. Potential evapotranspiration was also estimated over the same period using the Penman-Monteith formulation and then bias-corrected using the latter method. Both methods successfully corrected bi-

ases in a range of daily temperature, precipitation and potential evapotranspiration metrics, and reduced biases in multi-day precipitation metrics to a lesser degree. An exploratory analysis of the projected future changes confirms the expectation of wetter, warmer winters and hotter, drier summers, and shows uneven changes in different parts of the distributions of both temperature and precipitation. Both bias-correction methods preserved the climate change signal almost equally well, as well as the spread among the projected changes. The change factor method was used as a benchmark for precipitation,

and we show that it fails to capture changes in a range of variables, making it inadequate for most impact assessments. By comparing the differences between the two bias-correction methods and within the 12 ensemble members, we show that the uncertainty in future precipitation and temperature changes stemming from the climate model parameterisation far outweighs the uncertainty introduced by selecting one of these two bias-correction methods. We conclude by providing guidance on the use of the bias-corrected data sets. The data sets bias-adjusted with ISIMIP3BA are publicly available in the fol-

lowing repositories: https://doi.org/10.5281/zenodo.6337381 for precipitation and temperature (Reyniers et al., 2022a) and



https://doi.org/10.5281/zenodo.6320707 for potential evapotranspiration (Reyniers et al., 2022b). The datasets bias-corrected using the quantile mapping method are available at https://doi.org/10.5281/zenodo.8223024 (Zha et al., 2023).

## 1   Introduction

Climate model projections are essential to anticipate and adapt to future climate change impacts. Continuous efforts by the climate modelling community have led to major improvements in the realism of climate model simulations, yet significant discrepancies (biases) between observations and simulations of climatic variables remain (e.g. Kotlarski et al., 2014; Vautard et al., 2021). Biases in climate projections require particular attention when the projections are used to force impact models, for instance to assess future impacts on river streamflow, ecosystems or agricultural yields. Capturing the whole distribution of rainfall amounts is an essential prerequisite for hydrological modelling but is still notoriously challenging for many climate models. The response of impact models to forcing errors can be non-linear and amplify the severity of the biases, hence biases are typically adjusted before the climate projections are used in impact models.

To this effect, a range of bias-correction (BC) methods have been developed and compared (Teutschbein and Seibert, 2012; Gutmann et al., 2014; Maraun et al., 2019). These methods essentially transform the simulations so that some of their statistical properties match those of the observations. This efficiently reduces biases and, as a result, can considerably improve impact simulations (Rojas et al., 2011; Hakala et al., 2018; Pastén-Zapata et al., 2020). We note, however, that residual biases remain after the correction, and can deteriorate the impact simulations (Teng et al., 2015). This highlights that these biases are not corrected and removed, but rather, adjusted - as such, the term 'bias adjustment' is more accurate and becoming more widely used, but here we use BC to match how these methods are more commonly described in the literature. In addition, the statistical nature of BC methods means they only address the symptoms and not the origin of model errors, i.e. they do not identify the causes of model biases nor account for them (Addor et al., 2016; Maraun et al., 2017).

Furthermore, the reliability of BC can be questioned because of its reliance on the assumption that climate model biases are stationary in time or under a changing climate state (Maraun, 2012; Ehret et al., 2012; Teutschbein and Seibert, 2012; Chen et al., 2015; Hui et al., 2020). A related issue is that BC can modify the simulated climate change trends. This poses an issue if the origin of model errors in daily variability in an evaluation period (to which BC is often calibrated) differs from the origin of potential errors in the model's climate change response (Maraun et al., 2017). These challenges are difficult to overcome. However, when following the alternative approach of using unadjusted climate model output, propagation of biases through impact models can severely bias the resulting simulations and sometimes render them unusable (Hakala et al., 2020). As such, BC is usually necessary and widely used to assess climate change impacts, despite its imperfections.

In the context of the UK, several recent studies have been published on impact assessment by using the UK Climate Projections 2018 (UKCP18) regional climate model (RCM) 12km regional perturbed physics ensemble (UKCP18-RCM-PPE, referred to as UKCP18-RCM hereinafter, Met Office Hadley Centre, 2018; Lowe et al., 2018). The studies span various sectors including the impacts assessment on future river and groundwater flow (Kay et al., 2020; Hannaford et al., 2023; Kay et al., 2023), extremes (Hanlon et al., 2021) such as flooding (Griffin et al., 2022; Gudde et al., in review) and drought (Reyniers





et al., 2023), heat stress (Kennedy-Asser et al., 2022), and multi-sectoral analysis (Arnell et al., 2021). Although the UKCP18

data has been widely used, the UKCP18 project itself does not provide bias-corrected climatic variables for impact assessment. The aforementioned studies use different bias-correction methods for their relevant climatic variables, which potentially make the impact assessment less comparable due to methodological differences in bias-correction. The recently published CHESS-SCAPE dataset (Robinson et al., 2023) also provides UKCP18 bias-corrected data. It contains 11 near-surface meteorological variables, is downscaled to a 1 km spatial resolution, and is provided at several temporal resolutions (from daily to decadal

means) for the time period 1980–2080. For the temperature in the CHESS-SCAPE dataset, the seasonal (DJF, MAM, JJA and SON) offsets were calculated for each season and each 1km grid cell as the difference between CHESS-SCAPE and CHESS-met observations (Robinson et al., 2020a). The offsets were then subtracted from the CHESS-SCAPE data. For precipitation, the seasonal scaling factors were calculated as the ratio of CHESS-SCAPE to CHESS-met, and the CHESS-SCAPE precipitation were then multiplied by these factors. In the CHESS-SCAPE dataset, 4 out of the 12 RCM-PPE members (01, 04, 05

and 15) are provided. The bias-corrected CHESS-SCAPE data were shown to be consistent with the CHESS-met observations in the historical period 1980-2000. However, the simple BC methods used in the CHESS-SCAPE do not take into account changes in higher-order moments than the mean. Several studies have shown that while most BC methods are effective in correcting the means, the quantile mapping (QM) method outperforms simpler BC methods (offsetting or scaling) for temperature and precipitation for different statistical quantities including the standard deviation and percentiles (Teutschbein and Seibert,

2012; Fang et al., 2015; Azmat et al., 2018; Worku et al., 2020). The bias-corrected regional UKCP18 data produced as part of the enhanced Future Flows and Groundwater (eFLaG Hannaford et al., 2023) project have also been made available (Lane, 2022). Unlike CHESS-SCAPE, they are based on the full 12-member ensemble but they only include one variable, available precipitation (the sum of rainfall and snowmelt derived using snow module). The inclusion of snowmelt means that this is not directly comparable to the bias-corrected rainfall data introduced here, even though the raw rainfall data is the same. But is

worth noting that eFLAG relies on bias-correction factors computed from monthly means, i.e. based on the assumption that the whole distribution is affected by the same bias as the mean. We illustrate below that this assumption is often violated, e.g. the bias in the 95th percentile can be quite different from the bias in mean precipitation (Fig. 1). Furthermore, our understanding is that eFLAG only bias-corrected precipitation, and not temperature or potential evapotranspiration.

In this study, the precipitation and temperature of the UKCP18-RCM are evaluated and bias-corrected. The empirical QM

method and a change-preserving variant of the QM method were applied to bias-correct the UKCP18-RCM precipitation and temperature variables from all the 12 RCM-PPE members over the period of 1980-2080. In addition, potential evapotranspiration (PET) was computed from these UKCP18 simulations and bias-corrected using the trend-preserving BC method. The raw precipitation and temperature simulations and derived PET data were evaluated before the two BC methods were applied. The resulting bias-corrected datasets were evaluated and compared, and recommendation was made in the use of the datasets.

Specifically, the following questions are investigated:

1.  How biased are the UKCP18-RCM projections?

2.  Can existing bias-correction methods successfully correct errors in simple and more challenging metrics?



3. What climatic changes do the UKCP18-RCM ensemble members broadly project for the UK, and do the chosen bias-correction techniques affect them?

An exploratory analysis of the changes projected by UKCP18-RCM is discussed in section 3.2, using metrics based on daily precipitation and daily average temperature.

## 2   Data and Methods

### 2.1   Data

#### 2.1.1   The UKCP18 regional climate projections

The UK Climate Projections 2018 (UKCP18) are the current generation of national climate projections for the UK, developed by the Met Office Hadley Centre as part of their Climate Programme (Met Office Hadley Centre, 2018). The research presented in this study makes use of its third strand, an ensemble of the 12km regional climate model projections (UKCP18-RCM; Met Office Hadley Centre (2018), available from the Centre for Environmental Data Analysis (CEDA). The RCM simulations were run over the EURO-CORDEX rotated pole grid with a horizontal resolution of 0.11°, which results in quasi-uniform 12 km
spacing over the European domain (Murphy et al., 2018), and made available for the UK region with these coordinates as well as the OSGB36 projection which was used for the work presented in this thesis. Simulations of different variables are available from 1 December 1980 to 30 November 2080 on a daily time step (for practical reasons, December 1980 was left out of our analyses). The UKCP18-RCM simulations are a perturbed physics ensemble (PPE), obtained by running the global climate model (GCM) HadGEM3-GC3.05 with perturbations in 47 parameters from the convection, gravity wave drag, boundary layer,
cloud, large-scale precipitation, aerosols, and land surface interaction schemes. The global simulations were then downscaled by one-way nesting with a regional configuration of this model using the same perturbed parameter sets as its driving GCM ensemble member. The HadGEM3-GC3.05 parameter sets chosen for the global and regional UKCP18 PPE ensembles were selected in multiple stages based on different criteria, as summarised briefly below and explained in Murphy et al. (2018) and references therein. The UKCP18-RCM simulations were forced with the RCP8.5 scenario, a very high emissions scenario
characterised by high population growth and energy demand (Riahi et al., 2011).

#### 2.1.2   Observation data

As observational reference for the evaluation and bias-correction of UKCP18-RCM precipitation and temperature, the Met Office's 1 km HadUK-Grid dataset (Hollis et al., 2019) was used after regridding to the UKCP18-RCM 12 km grid by averaging all 1km grid points that lay in each UKCP18-RCM grid cell (consistent with Hollis et al., 2019). For the bias-correction and
evaluation of PET, the CHESS-PE dataset provided by the Centre for Ecology and Hydrology was used (Robinson et al., 2020b), also after regridding by averaging to the UKCP18-RCM 12 km grid. Daily data from 1981 to 2010 were used here (December 1980 was omitted for practical reasons).





## 2.2 Potential evapotranspiration

Potential evapotranspiration (PET) is not provided by UKCP18-RCM but it is a necessary input variable for some impact
models and indicators, so it was calculated off-line here. While rising temperatures lead to PET increases, changes in humidity,
net radiation and wind speed can also play a significant role. Therefore, PET was calculated using the Penman-Monteith
method, which includes the effect of all these variables and is recommended over simpler temperature-based methods (e.g.
Dewes et al., 2017), although it is still subject to significant limitations (Milly and Dunne, 2016; Greve et al., 2019). The
calculation of PET for the UKCP18-RCM used here relies on the same variant of the Penman-Monteith method used by
Robinson et al. (2017), to ensure consistency with the CHESS-PE dataset. Specifically, the following variables simulated by
the UKCP18-RCM ensemble were used: specific humidity, pressure at sea level, net downwelling longwave radiation, net
downwelling shortwave radiation, wind speed at 10m and daily average surface air temperature. PET was set to zero wherever
a calculated value was negative (which occurred for less than 1% of the values overall and, when split by ensemble member
and month, also less than 1% for all cases except December in ensemble member 1 with 1.2% of negative values).

## 2.3 Evaluation and trend analysis


Biases in UKCP18-RCM precipitation and temperature metrics were assessed over the reference period 1981-2010 (REF),
using the 30-year temporal averages of a range of metrics computed for each grid cell. For analysing projected changes, the
final 30-year period of the simulations was chosen as the future period (2051-2080; FUT) for each ensemble member.

Climate model errors are not necessarily homogeneous across the range of the precipitation and temperature distributions.
Therefore, the initial model evaluation metrics shown consist of the errors in the mean, in a lower tail metric (Q05 for tem-
perature and PET, dry day frequency for precipitation), and in Q95 as an upper tail metric. This is based on the results of a
preliminary analysis.

Changes in temperature and precipitation extremes are generally of greater societal interest than changes in the mean (al-
though extreme impacts do not always need extreme meteorological conditions to arise; van der Wiel et al., 2020). Therefore,
a set of moderate and extreme climate indicators was used to further evaluate the model error and analyse projected changes of
simulated precipitation and temperature. These metrics were drawn from or inspired by the list of indices compiled by the Ex-
pert Team on Climate Change Detection and Indices (ETCCDI; http://etccdi.pacificclimate.org/list_27_indices.shtml), which
have been extensively used in the literature, including in IPCC reports (IPCC, 2021). Daily mean temperature was used in this
study, so the ETCCDI temperature indicators (which typically use daily minimum and maximum temperatures) were modified
to use mean temperature. Table 1 gives an overview of all indices used and their definitions.

## 2.4 bias-correction

Comparison to observations revealed significant biases in the simulations of precipitation, temperature and PET, so these
variables were statistically post-processed. Two closely related BC methods (quantile mapping and the ISIMIP3BA approach,
see below) were used to bias adjust UKCP18-RCM, to allow exploring the sensitivity of the results to the differences between



**Table 1.** Climatic indices used to evaluate and examine trends in precipitation, temperature and PET simulated by UKCP18 and derived PET.

| Index | Description |
| --- | --- |
| *Temperature* | |
| yMEAN | Average annual (seasonal) mean daily-mean temperature (°C) |
| yMIN | Average annual (seasonal) minimum daily-mean temperature (°C) |
| yMAX | Average annual (seasonal) maximum daily-mean temperature (°C) |
| Qn | Daily-mean temperature exceeded on average n% of days in a year (season) (°C) |
| *Precipitation* | |
| prTOT | Average annual (seasonal) total precipitation (mm) |
| SDII | Simple precipitation intensity index: average annual (seasonal) mean wet-day precipitation intensity (mm) |
| DF | Average annual (seasonal) dry-day fraction (%) |
| Qn | Daily precipitation exceeded on average n% of days in a year (season) (mm) |
| Rx5day | Average annual (seasonal) maximum five-day total precipitation (mm) |
| CDD | Average annual (seasonal) maximum number of consecutive dry days (days) |
| CWD | Average annual (seasonal) maximum number of consecutive wet days (days) |
| *Potential Evapotranspiration* | |
| Qn | Daily PET exceeded on average n% of days in a year (season) (mm) |
| mean | Mean annual (seasonal) PET (mm) |

the BC methods, including but not limited to whether or not the BC method explicitly aims to preserve the climate changes. Both BC methods were applied to each grid cell, ensemble member and calendar month combination separately, and were calibrated using simulated and observed data for 1981–2010. For precipitation, we also compared future projections from these BC methods with the change factor method, which is not considered a bias-correction method but consists of applying projected changes in mean climate to observed time series. The change factor method was applied using the same period as the 155 BC methods.

### 2.4.1 Quantile Mapping

Quantile mapping (QM; e.g. Piani et al., 2010) is a statistical transformation of the distribution of a modelled variable such that it matches the distribution of the observed variable. By construction, the resulting distributions of the reference period simulations and observations match closely, removing deviations from the observed data in the mean, variance and higher order 160 moments. The application of the mapping to either the quantiles of an empirical cumulative distribution function (CDF) or an assumed parametric CDF can result in varying performance of the QMs, depening on the transformation functions employed



(Enayati et al., 2021). Switanek et al. (2017) demonstrated that, where the parametric distribution is known to be a perfect fit to the observed and simulated data (because, e.g., it is synthetic data drawn from that distribution), then parametric QM reduces the influence of sample size-induced noise that degrades the empirical CDFs of observed and simulated time series. However,
Gudmundsson et al. (2012) found that empirical BC methods were more successful at bias-correcting different precipitation quantiles than fitted distributions or simpler parametric methods.

Here, the choice was made to use non-parametric QM with an empirical distribution, as a complement to the second BC method which uses a parametric distribution. The resulting dataset will be refered to as BCQM. To implement QM, the R package qmap (Gudmundsson et al., 2012) was used, with QM applied to 1000 empirical quantiles. The type of interpolation
used between the fitted transformed values is linear interpolation. It is worth noting that Lafon et al. (2013) demonstrated error reduction for the empirical distribution method increases with the number of quantiles used. They have done a cross-validation test of empirical QM with 25, 50, 75, and 100 quantiles, and 100 quantiles vielded the best results, excluding any cases with over 100 quantiles. Our study expanded the number of quantiles to include 1000 for comparison with the ISIMIP3BA BC method. However, it should be acknowledged that increasing the number of quantiles may potentially compromise the efficiency or lead
to overfitting issues. Our analysis shows that the differences between this non-parametric method and the parametric method introduced below are overall minor, implying that in this case the climate signal is not particularly sensitive to any overfitting that may have resulted from this, especially when compared to other decisions such as the climate model selection. QM was not used to bias adjust PET.

### 2.4.2   ISIMIP3b bias-correction method

The second method used in this study is the univariate change preserving bias-correction method developed for phase 3b of the Inter-Sectorial Impact Model Intercomparison Project (BCI3;  ) Lange (2019). A parametric quantile mapping method which approximately preserves the climate change signal in each quantile was applied to each variable independently, with a different distribution used for each variable. Here, we corrected daily average temperature, precipitation and potential evapotranspiration using the normal, gamma and Weibull distributions, respectively. The choices for temperature and precipitation here were
motivated by their use in the literature and by Lange (2019), combined with verifying the results. For PET, the Weibull, gamma and beta distributions were fitted to sample grid cells representative of different climates across the UK. While all three would have been adequate, the Weibull distribution was used in the end due to slightly better performance in the sample cells, and, using the Kolmogorov-Smirnov test statistic, it showed a better fit than the beta distribution in most regions of the UK in most months.

For precipitation and potential evapotranspiration, the BCI3-option to apply a separate correction to the probability of occurrence of events beyond thresholds was used to adjust bias and preserve the projected change in the frequency of dry days / no PET ($< 0.1$ mm/day). For each combination of ensemble member, month and location, the change in the distributions of precipitation, PET and temperature between present-day (calibration) and future periods was preserved by computing the change for 50 quantiles and applying it to the observed quantiles. This was done multiplicatively (but additively if there are
large negative biases in the model data, to avoid obtaining unrealistically large values; Lange, 2019) for precipitation and PET,





and purely additively for temperature. For PET and temperature , the transient trend was preserved by removing it before bias-correction and adding it back afterwards, as described in Lange (2019).

The code used for this bias-correction is version 2.4.1 of the Python code made available on Zenodo by Lange (2020). Version 2.4.1 differs from the originally published method (version 1.0; Lange, 2019) in the equation used for the correction 200 of the frequency of events beyond thresholds (in this case dry days) and code error fixes. The code was modified for this study in minor ways: to allow it to be used when simulations and observations have different calendars (360-day and Gregorian, respectively) and to suit the format of the data used.

The BCI3 method was applied to, and preserves changes between, two 30-year periods (the 1981-2010 reference period and future application periods). This can introduce artefacts such as step changes between adjacent future 30-year application 205 periods and imperfect preservation of the trend within each 30-year application period. In order to produce a continuous, bias-corrected 100-year sequence (1981-2080), the algorithm was applied to overlapping periods of 30 years (same length as the reference time period), with the starting point of the application period increasing in decadal steps. Then, the central 10 years (as well as the first and last 10 for the first and last 30-year period, respectively) of each run was extracted and concatenated to obtain the final semi-transient bias-corrected time series. Note that the climate change signal in the very first and last decades 210 of the resulting concatenated bias-corrected time series may be slightly stronger or weaker, respectively, because here the beginning and end of the application periods were used instead of the central decade. To estimate the magnitude of the effect of this strategy, the 10-year overlaps of pairs of 30-year bias-corrected periods separated by 20 years were used to examine differences in the resulting distributions. The decades 2001-2010 and 2051-2060 (i.e. the earliest and latest of the six decades that were part of three 30-year chunks, respectively) were chosen for making scatter plots and QQ-plots for precipitation and 215 temperature in two grid cells representing different UK climates. For each decade, the three sets of BC data are called 'tail', 'middle' and 'head', depending on whether they are the first, middle or last decade of the 30-year bias-correction period, respectively.

### 2.4.3 Change factor method

The change factor method is commonly used in impact studies (e.g. Prudhomme et al., 2012; Kay et al., 2020). It is included 220 here as a benchmark for precipitation only, for the purpose of demonstrating the limitations of basing impact studies on an observed time series perturbed only by a change in mean climate, and thus show the added value of assessing changes based instead on the RCM output itself. A multiplicative change factor ($CF$) was computed and applied to the simulated precipitation time series for each month $m$ as follows:

$$CF_{m,p} = \frac{\overline{P}_{raw,FUT,m,p}}{\overline{P}_{raw,REF,m,p}} \tag{1}$$

225 where $\overline{P}$ is the mean precipitation from the raw UKCP18-RCM data over the future ($FUT$) or calibration ($REF$, i.e. reference) period for month $m$ and ensemble member $p$. Each monthly $CF$ is then applied to the observed calibration period time series to generate the CF precipitation values for each timestep $i$ within month $m$:



$$P_{CF,i,p} = CF_{m,p} P_{obs,i} \tag{2}$$

## 3 Results

### 3.1 Evaluation

#### 3.1.1 Bias of raw simulations

The maps in Fig. 1 show the ensemble mean errors of the raw UKCP18-RCM projections in the dry-day frequency, mean daily precipitation and the Q95 of precipitation in the reference period, expressed as a percentage of the observed value. In general, the frequency of dry days in UKCP18-RCM is too low (and therefore the wet-day frequency is too high), particularly in the winter and in regions of higher elevation. In summer, the dry-day frequency bias is very small for most of England. The precipitation mean and Q95 are strongly overestimated across the UK in winter, although in highly elevated areas this bias is smaller or even reversed in sign (especially for Q95). In summer, however, the mean and Q95 biases show a strong spatial variability, with underestimations toward the south and at high elevation levels, and a wet bias in the north of the UK. These seasonal bias differences result in an annual bias of too few dry days almost everywhere, too wet mean precipitation in most regions, and more mixed wet and dry Q95 relative biases.

The maps in Fig. 2 show the ensemble mean errors of the raw UKCP18-RCM projections in the mean daily temperature as well as the cold (Q05) and hot (Q95) tails of the distribution. At the annual scale, temperature tends to be underestimated in UKCP18 across its distribution, as reflected by the mean, Q05 and Q95. Temperature biases in winter show a gradient from north to south, with generally cold biases in the north and warm in the south. Along the north-south axis, biases are generally colder in the lower tail and warmer in the upper tail of the distribution, indicating an overestimation of temperature variability in winter. In the middle regions along the north-south axis, the biases in the cold and hot tails of the distribution have opposite signs. UKCP18-RCM underestimates temperature on cold days (Q05) in the north especially strongly. In summer, temperature is typically underestimated across the UK in all three indices considered. One exception is the overestimation (or smaller cold bias) in Q95 in major built up areas (e.g. London in South East England). Biases in the representation of urban heat islands were previously documented by Lo et al. (2020), who found that UKCP18-RCM tends to overestimate the intensity of urban heat islands in summer, more so for nighttime than daytime.

The maps in Fig. 3 show the absolute UKCP18-derived PET biases in mm. Note that these maps do not show Northern Ireland, as this region is not included in the CHESS-PE dataset which served as the observational reference. The predominantly cool bias in the UKCP18-RCM ensemble mean (Fig. 2) contributes to the low PET bias in some regions and seasons (Fig. 3). Interestingly, however, the regional and seasonal variations in the PET biases do not closely follow the temperature bias patterns, implying that the bias in PET is not solely caused by errors in the daily average temperature.

Note that the strong positive temperature biases shown by some grid cells along the coast in winter come from the use of the regridded 1km HadUK-Grid, which covers the land only, and contrary to UKCP18-RCM does not account for the warmer

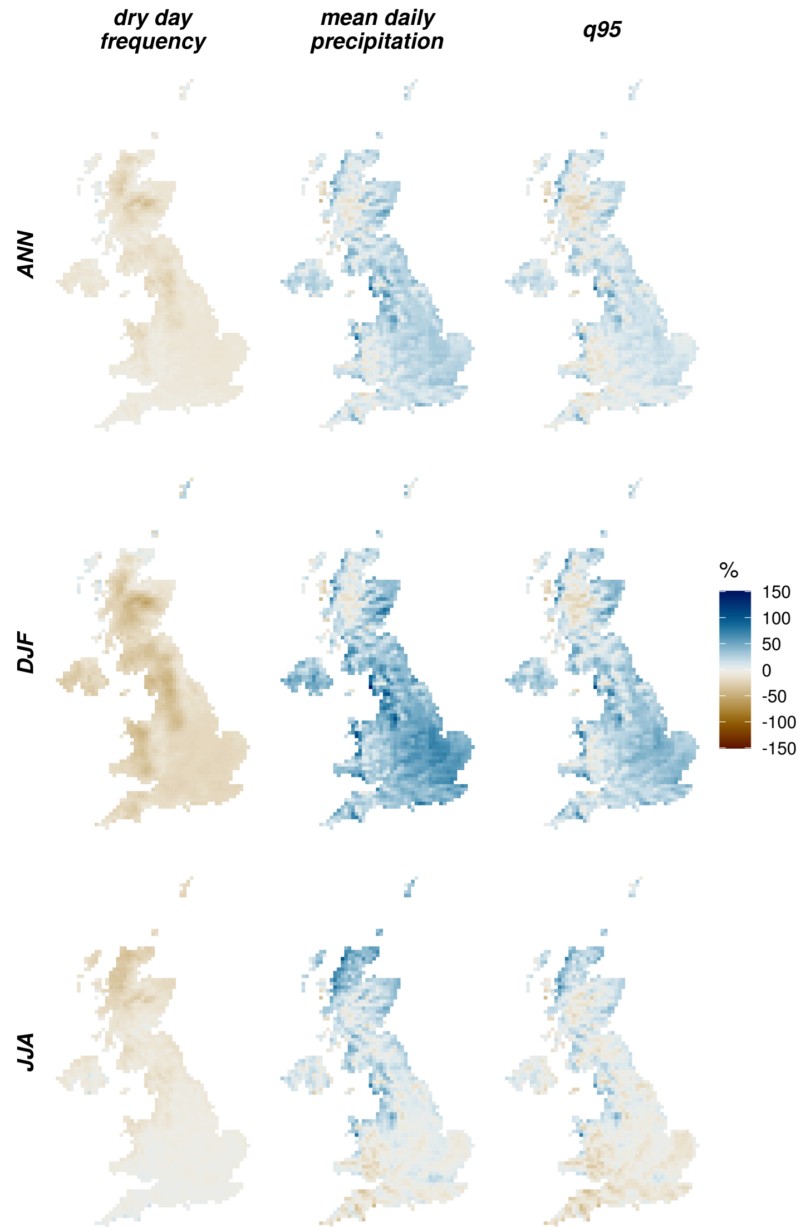

**Figure 1.** Precipitation biases in UKCP18-RCM for 1981-2010, expressed as a percentage of the observed values. The percentage bias for each ensemble member was computed and the mean across the ensemble is shown. Dry-day frequency is the percentage of days with P < 1 mm; mean daily precipitation is the precipitation averaged over all days; q95 is the 0.95 quantile of precipitation across all days. Top, middle and bottom rows are for annual, DJF (December, January, February) and JJA (June, July, August), respectively.

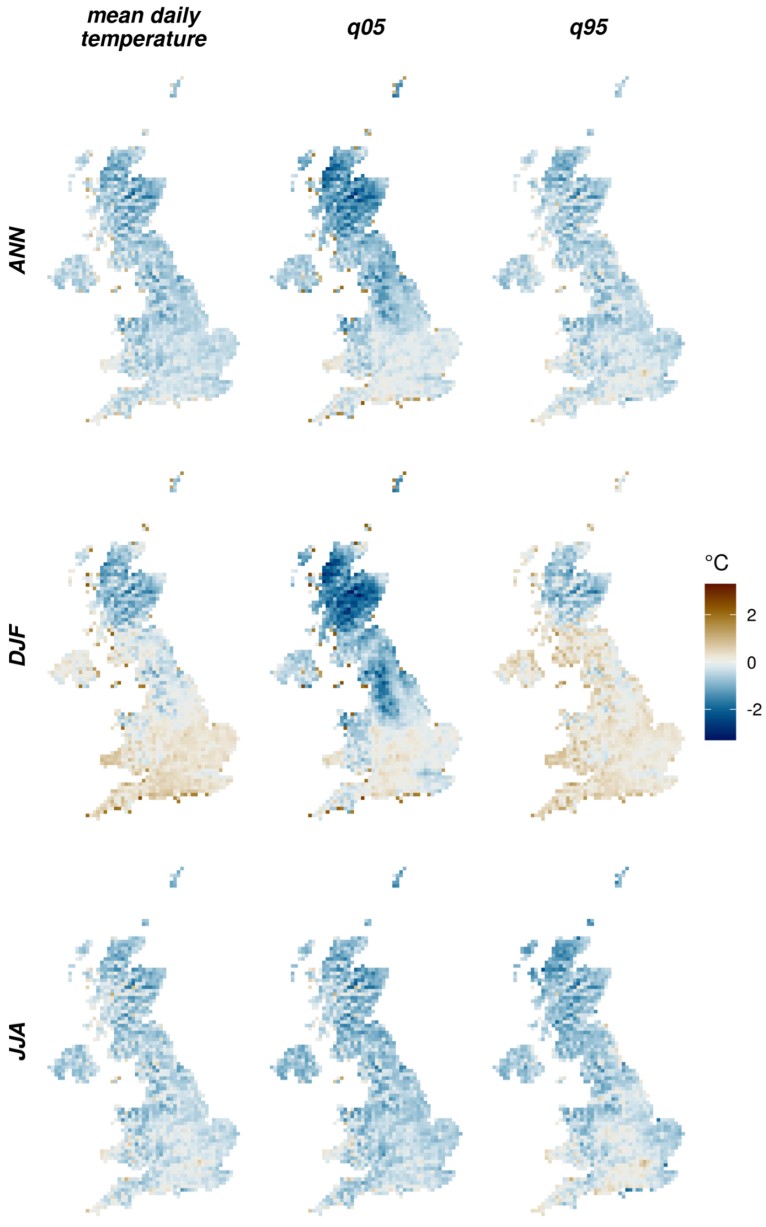

**Figure 2.** Temperature biases (°C) in UKCP18-RCM for 1981-2010. The bias for each ensemble member was computed and the mean across the ensemble is shown here. Q05 and Q95 are the 0.05 quantile and the 0.95 quantile across all days, respectively.





**Figure 3.** As Fig. 2 but for PET (mm/day).





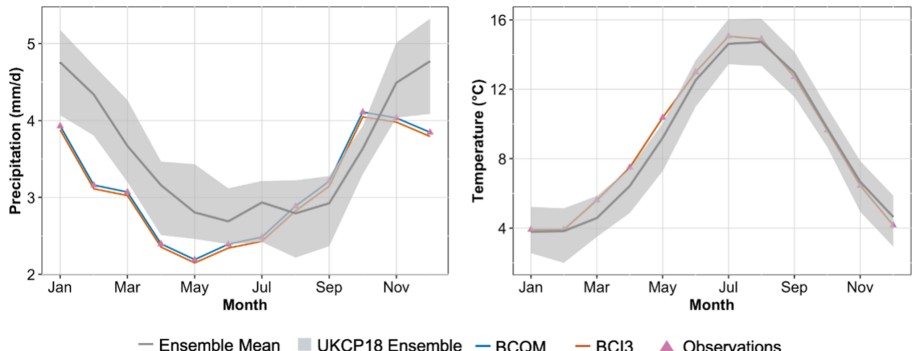

**Figure 4.** Comparison between observations and UKCP18-RCM simulations before and after bias-correction for monthly precipitation (left, mm/day) and temperature (right, °C) averaged over the UK for 1980-2010. The grey shading shows the spread of the 12-member ensemble, prior to bias-correction. The blue line lies underneath the red line in the temperature panel).

sea temperature in winter, leading to "observed" grid cell averages that are too low. The opposite but smaller effect occurs in
summer, and a similar effect can be observed in the PET bias maps.

Figure 4 shows the UK average seasonal cycle of the raw and bias-corrected precipitation and temperature. For precipitation, the raw UCKP18-RCM ensemble is too wet from November to June, and its range does not encompass the observations during these months. The seasonal timing also appears to be shifted: the driest and wettest months are delayed in the simulations (June and January) compared to the observations (May and October). The ensemble averaged daily mean temperature generally
matches observations far more closely than precipitation (although individual members can contain biases of either sign). The largest mean temperature biases over the UK occur from March to May, a period during which almost all members are too cold.

### 3.1.2 Evaluation of bias-correction

The biases of monthly mean temperature and precipitation are effectively corrected by both BCQM and BCI3, with observa-
tions very closely matching the processed ensemble members (Fig. 4). The remaining error in the statistics shown on the maps in Figures 1, 2 and 3 are small (and therefore not shown here, as they take up much space for little information).

Figures 5 and 6 show heatmaps of the spatially averaged errors in precipitation and daily mean temperature respectively, before and after bias-correction. The value of each metric was calculated for each RCM ensemble member and for the observations, at each grid cell for each year and then averaged over 1981-2010. The heatmaps show the spatial average of the absolute
values of the difference between observation and simulation (as a percentage of the observed value for precipitation). For the metrics which are also shown on the maps in Figures 1 and 2, the UK-averaged errors are strongly reduced for the full year as well as winter and summer for all ensemble members, resulting in a large decrease in the ensemble mean errors. The standard deviation (sd, bottom rows) of the errors decreases largely as well, as the statistics of the ensemble members converge toward those of the observations for the reference period.



A close look at the figures in Fig. 5 and 6 reveals that biases may be smaller in BCQM than in BCI3, which is not surprising, since empirical quantiles were fitted for BCQM while BCI3 relied on parametric distribution. This should not be interpreted as an advantage of BCQM over BCI3, as the slight edge conferred by the non-parametric fit is likely to be lost when the bias-correction is applied to a period not used for calibration (Chen et al., 2019).

      For precipitation, BCQM outperforms BCI3 for reducing the bias in the total precipitation (prTOT) (Fig. 5). Since both

methods perform very well for correcting the DF and Q95 bias, the difference in skill for prTOT may be related to the wet end ($>$ Q95) of the precipitation distribution: BCQM (with 1000 quantiles) adjusts the simulated extremes more precisely to fit the observed extremes, while the fit of the gamma distribution used in BCI3 can deviate more from the observations for those extremes. Note that this is not necessarily a disadvantage in BCI3. BCQM may be overfitting to the most extreme events occurring in the 30-year observations(Switanek et al., 2017). Due to the strongly skewed nature of precipitation distributions,

its extremes (in this case, the upper tail beyond Q95) make a large contribution to the mean (Pendergrass and Knutti, 2018). Evaluation based on the mean may thus be influenced by how well these extreme quantiles are reproduced. In reverse, this reasoning implies that bias-correction relying solely on errors and projected changes in the precipitation mean could be sensitive to the variability in observed and projected extreme events, including relatively rare events that might be included or excluded depending on the choice of calibration period.

Although biases in daily precipitation statistics (TOT, DF, SDII) are largely removed, biases in multi-day metrics (CWD, CDD and Rx5day) still remain although reduced. This is expected (Addor and Seibert, 2014), as both BC methods are designed to correct biases in daily values, but not the temporal structure of the time series. For instance, although the dry day frequency is well corrected by both BC methods, these methods fail to correct errors in the sequences of consecutive wet days in winter and on an annual basis. This is due to an over-correction, from mostly overestimated toward the east to mostly underestimated

toward the west, where the observed CWD is longer. On the contrary, both methods reduced the summer CWD and the CDD in each time period by about half on average.

      For temperature, the biases in the mean are almost entirely removed by both methods (Fig. 6). The ensemble mean, Q05 and Q95 biases are strongly reduced, by a factor 7 to 18 for the year and by a factor 3 to 7 for winter and summer. The most persistent biases are found in the winter (and annual) minimum daily temperature (computed for each season and then averaged

over the whole period). The remaining winter maximum temperature bias is also markedly higher using BCI3 (about half the raw bias on average) than using BCQM (less than a quarter).

      Importantly, across the metrics in Figures 5 and 6, the differences between BCQM and BCI3 are generally relatively modest during the reference period. Both methods effectively reduce biases in single-day metrics, and they mostly struggle with the same metrics, and to some extent, with the same ensemble members.

To conclude this evaluation, Figures 7 and 8 assess whether errors were induced by concatenating the centre period of 30-year time series to make BCI3. These figures only show the results for ensemble member 1, as the results were similar across the ensemble. The 'middle' daily values (and quantiles for the QQ-plots) are plotted on the x-axis, and the corresponding 'tail' and 'head' values (quantiles) are plotted on the y-axis. For temperature, the uncertainty introduced by using different periods for change preservation is very small. For precipitation, the match between the quantiles is good (dark circles), although there



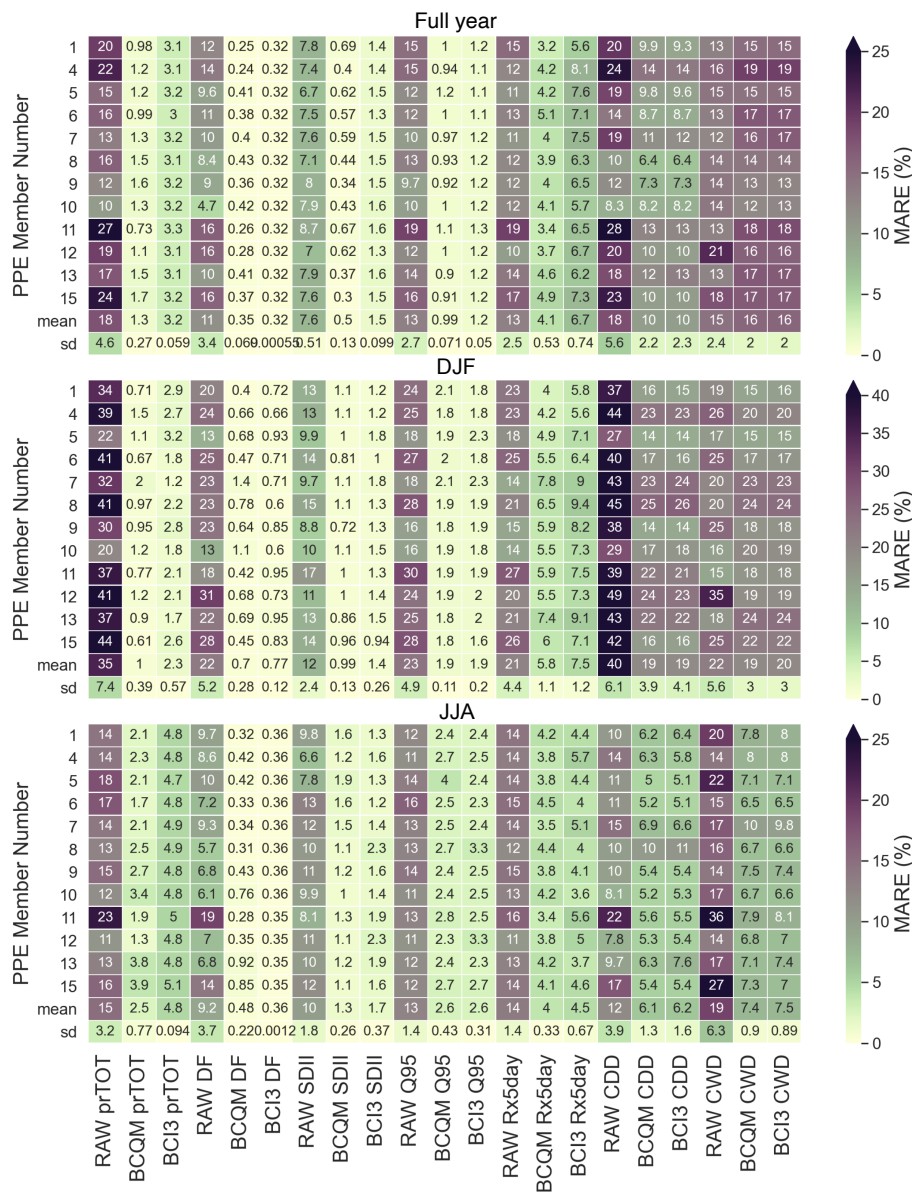

**Figure 5.** Mean absolute relative error (MARE, %) for calibration period precipitation, for the UKCP18 RCM simulations before (RAW) and after bias-correction (BCQM and BCI3). Values shown are UK-wide averages for 1981-2010. prTOT = annual total precipitation, DF = fraction of days that are dry (P < 1 mm), SDII = mean wet-day precipitation, Q95 = 0.95 quantile of daily precipitation, Rx5day = maximum 5 consecutive-day precipitation, CDD (CWD) = maximum number of consecutive dry (wet) days. The mean and standard deviation across the 12-member RCM ensemble are included at the bottom of each panel.



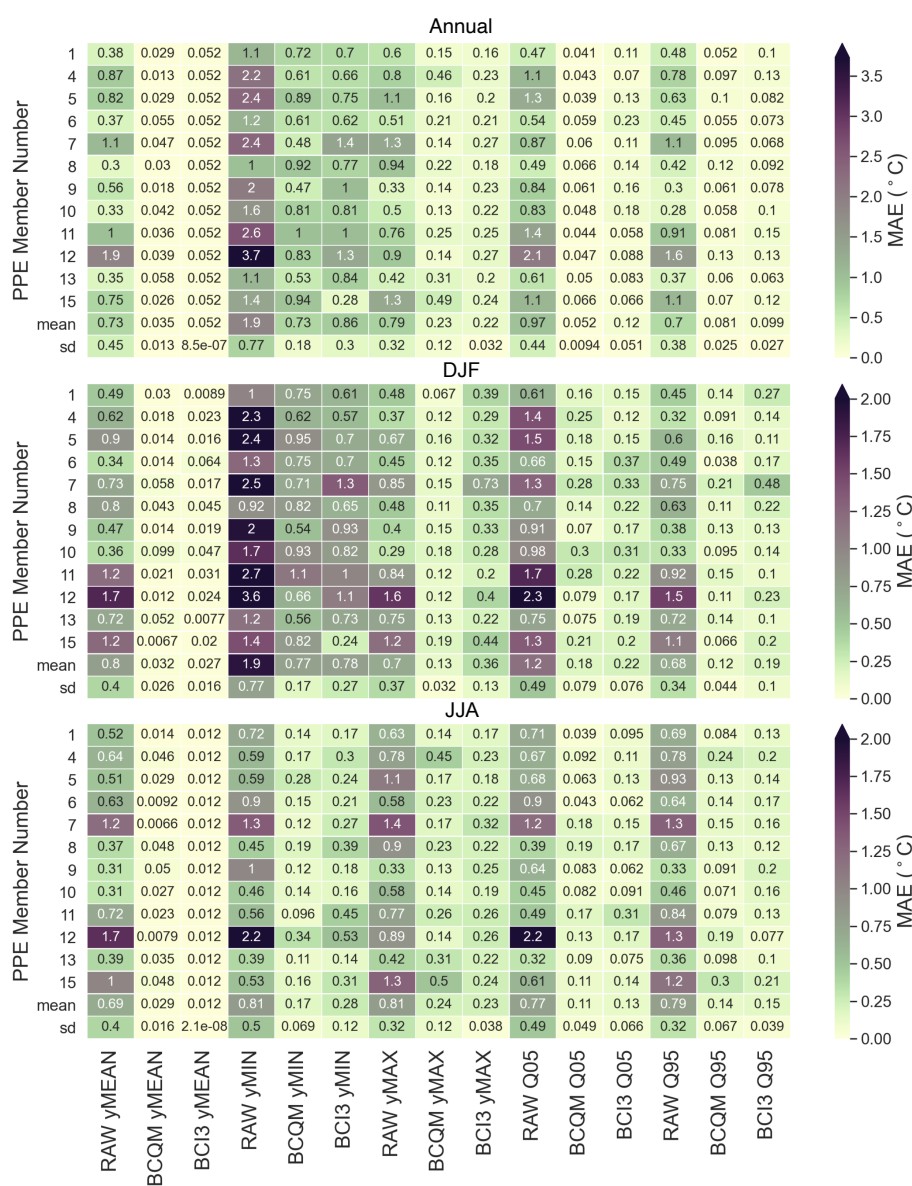

**Figure 6.** As Fig. 5 but for temperature, showing the bias (mean absolute error; MAE; °C) of the mean (MEAN), minimum (MIN), maximum (MAX), 0.05 quantile (Q05) and 0.95 quantile (Q95) of daily temperatures over each year (top), winter (middle) or summer (bottom).



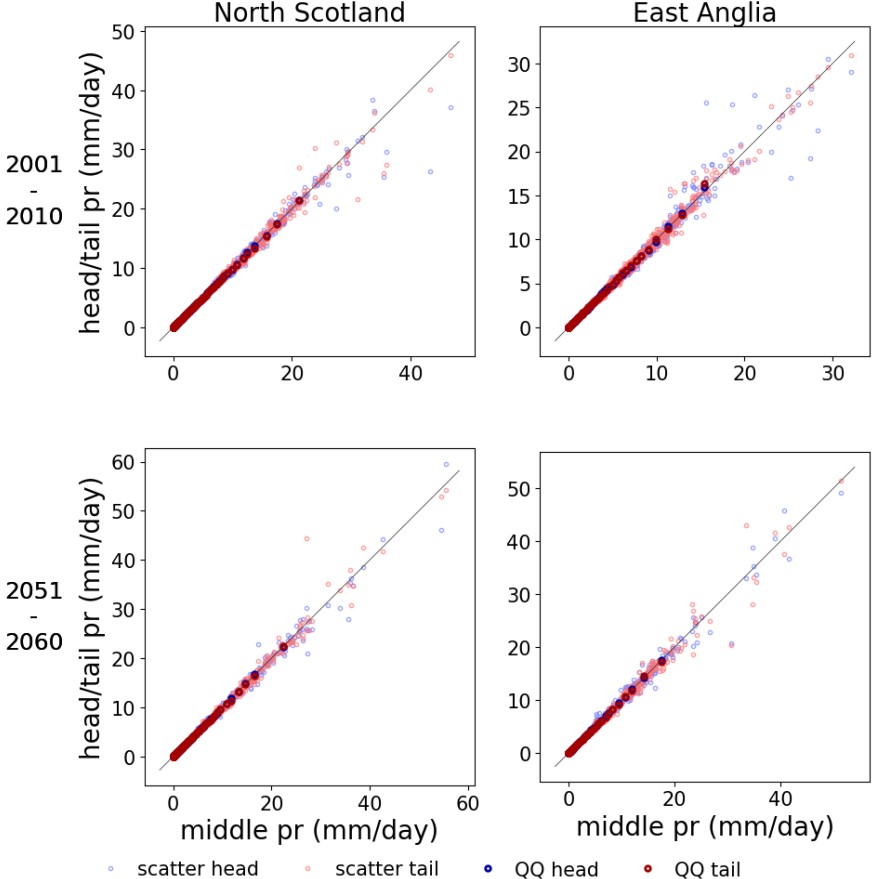

**Figure 7.** Quantile-quantile plot (dark colours) and scatter plot (light colours) of precipitation (pr) in two overlapping decades (top row: 2001-2010, bottom row: 2051-2060) for UKCP18-RCM ensemble member 1 for two example grid cells (columns). Tail, middle and head respectively refer to the first, middle and end 10 years of each 30-year period that overlaps in these decades. The line $y=x$ is shown for reference.

is some uncertainty introduced for the most intense precipitation events (light circles). The precipitation range covered by these

dots in the upper-right corner of the panels highlights the variability within the wettest quantiles, highlighting that when fitting

the upper tail of a 30-year period distribution, it is unlikely to perfectly match the upper tail from another period, in particular

in presence of climate change. Since some large differences in extreme rainfall can be introduced depending on the period

used, the concatenated time series or the data from each 30y period can be used when the focus is on these events, in order to

capture the influence of the bias-correction. Overall, the differences appear to be small enough to justify concatenation.

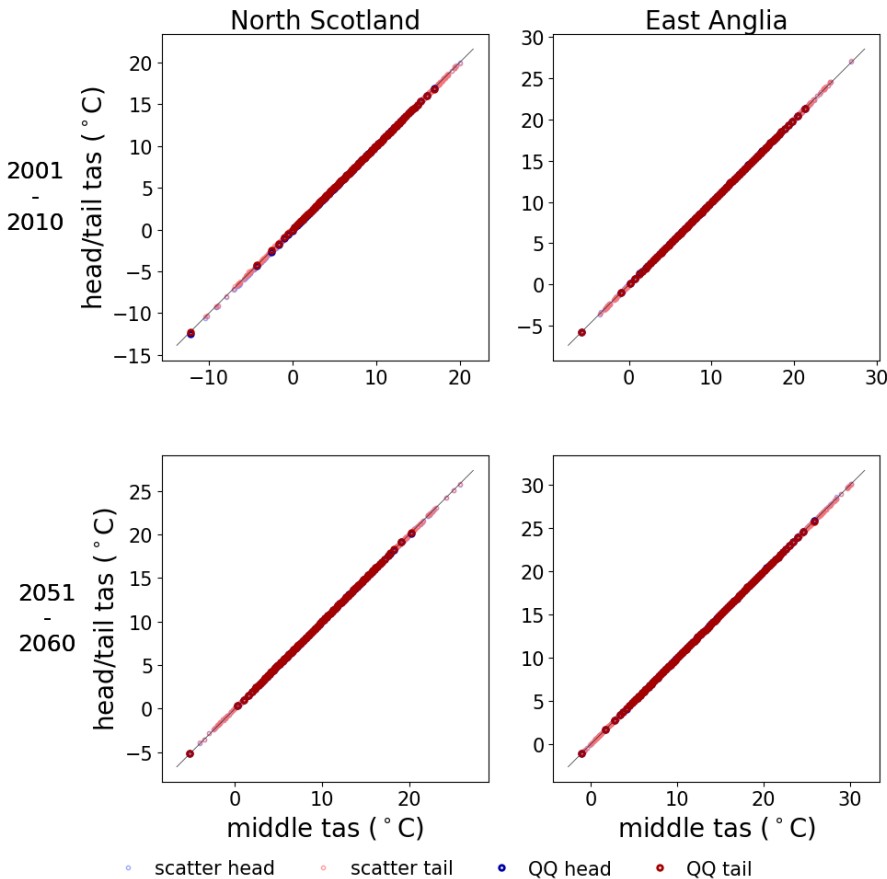

**Figure 8.** As Fig. 7 but for temperature.

## 3.2 Projected changes before and after bias-correction

This section analyses the climate change signal (CCS) projected by UKCP18-RCM in the evaluation metrics used above (Section 3.1), and particularly considers whether BCQM and the change-preserving BCI3 alter the CCS compared to the raw simulations. Furthermore, for precipitation, these changes are compared to those simulated using the change factor approach. The CCS is defined as the relative or absolute difference between metrics computed over 1981-2010 and 2051-2080. Maps of projected changes in the mean precipitation and temperature (Figures 9 and 10) show the spatial variability of the CCS in UKCP18-RCM mean, while the CCS aggregated over the whole domain is summarised for each member and index in heat maps (Figures 11 and 12).

The CCS of mean precipitation and temperature in the raw projections (left columns in Figures 9 and 10) confirms the conclusions of the UKCP18 headline findings: warmer, wetter winters and hotter, drier summers. The precipitation CCS shows substantial regional variation in all seasons, influenced by topography. The summer drying signal is strongest toward South





West England, while the winter wetting signal is weakest in the north-east of Scotland and strongest along the west and south coasts. The seasonally contrasting changes result in an overall precipitation decrease over most of the UK, except along the west coast of north England and Scotland. For temperature, the annual and winter projections are more spatially homogeneous than the summer projections, which show an increasing gradient from the north-west toward the south and south-east.

Overall, these regional variations in the projected changes in mean annual, winter and summer precipitation and temperature over the UK are well preserved in both BCQM and BCI3. In BCQM, the summer temperature increase contains more local variation and is somewhat exacerbated over the regions of higher elevation, and the winter precipitation increase is slightly greater than in the original projections. Spatial patterns in the local variations in the BCQM projected summer temperature changes resemble those in the summer temperature errors (Fig. 2). The raw changes are better preserved in BCI3, although the added value of this method in terms of change preservation is limited.

The heatmaps in Figures 11 and 12 summarize the UK-averaged projected changes in precipitation and temperature metrics, and allow to compare the contributions of i) the different ensemble members and ii) the two bias-correction techniques to the uncertainty in the projected changes. Before proceeding to a discussion of the effects of the different bias-correction techniques, the projected changes are briefly discussed below (based on the columns labelled 'RAW ...' in the heatmaps).

More so than the mean, precipitation variability has a profound influence on society, and how it is influenced by climate change is thus of great interest. In general, the 'wet' precipitation metrics show wetter (or less dry) projected changes than the total precipitation. The projected total precipitation decrease for summer is the combined effect of an increasing proportion of dry days (DF), and slightly decreasing precipitation falling on wet days, marked by greater relative decreases in the Q95 than in the SDII. The longest summer dry spell is projected to lengthen by a third on average (with wide variation across the ensemble). In winter, the relative increases in prTOT can be attributed to similar increases in SDII Q95 and Rx5day, rather than changes in the fraction of wet days. Interestingly, this is paired with a slight shortening of the longest wet day streak on average. Annually, these seasonal changes combine into a slight projected prTOT decrease and a modest increase in the fraction of dry days, combined with wetter wet days, including in the wet end of the precipitation distribution. These findings are in agreement with increases in moderate- and high-impact 1-day rainfall threshold exceedances found by Hanlon et al. (2021).

The projected average temperature increases in the UK are greater in summer than in winter. Projected changes in the range of daily mean temperatures are opposite between winter and summer: in summer the Q95 and maximum daily mean temperatures increase more than the Q05 and minimum daily mean temperature, whereas in winter the colder end of the tail is projected to warm more than the warmer end of the tail. There exists a partial overlap between the ensemble members with high summer CDD increases (6, 9, 11 and 13), and the ensemble members with the highest maximum daily-mean temperature increases (5, 8, 9, 11, 13). In line with the projected increases in minimum temperatures, Hanlon et al. (2021) found a continued decreasing number of days with minimum and maximum temperatures below 0°C (frost and icing days respectively). On the warm end of the distribution, the larger increases in the expected maximum mean daily temperatures match their findings of increasing days with maximum temperatures exceeding 25°C, and increasing nights with minimum temperatures over 20°C. Similarly, Arnell et al. (2021) found increases in the annual probability of experiencing at least one heat wave (based on regionally varying thresholds).

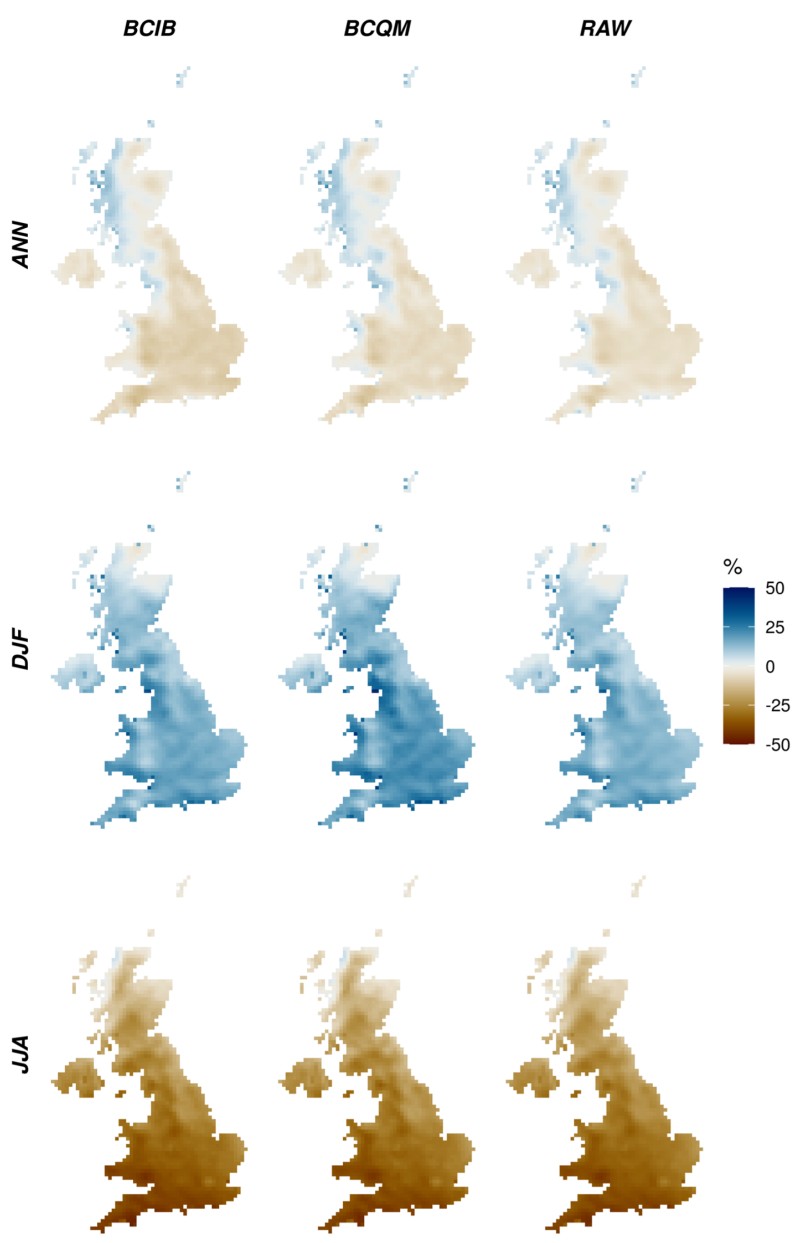

**Figure 9.** Projected changes in annual or seasonal mean precipitation from the ensemble mean of UKCP18-RCM simulations before (RAW) and after bias-correction (BCQM and BCI3). Values shown are the percentage change from 1981-2010 to 2051-2080 under RCP8.5.



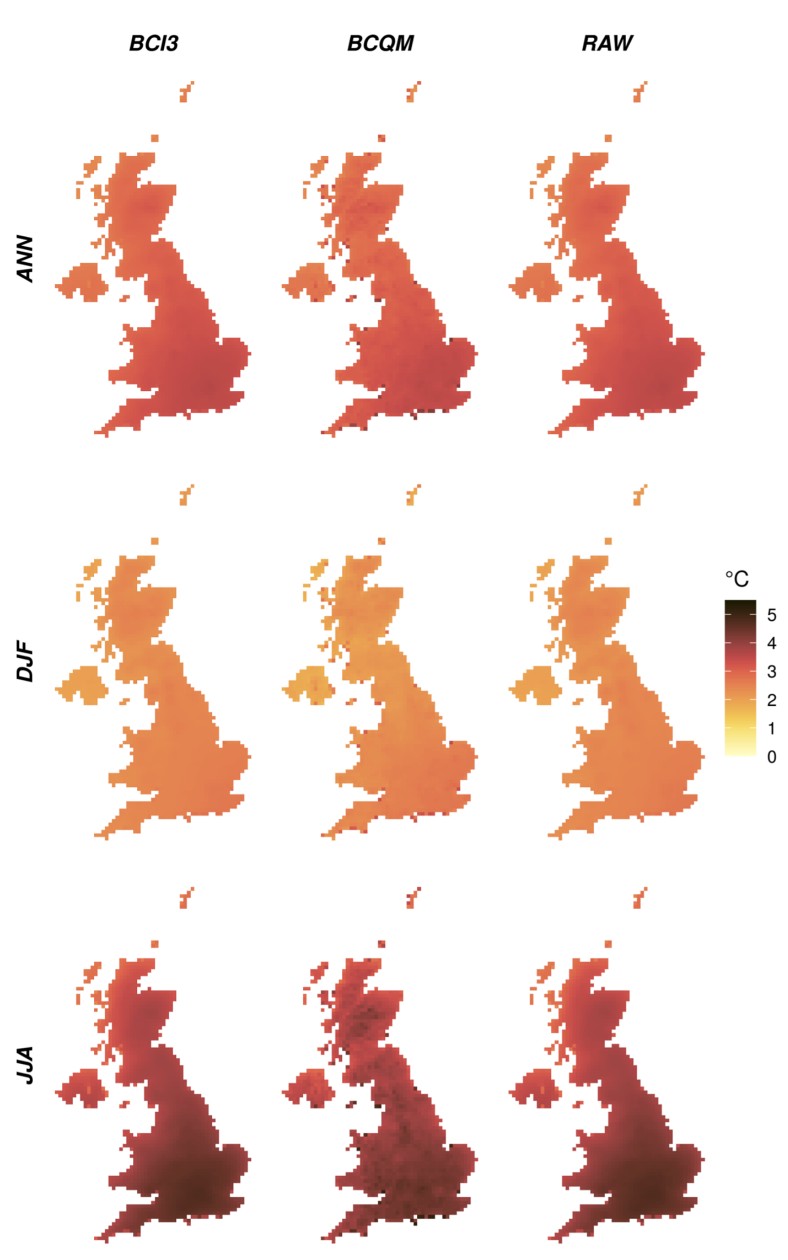

**Figure 10.** As Fig. 9, but for the change in temperature expressed in °C.





The differences between the rows for the different metrics and columns for bias-correction methods in the heatmaps (Figures 11 and 12) show a wide range of projected changes among the ensemble members, which usually exceeds the differences from BC method. In other words, at least when aggregated over the UK, the differences between the ensemble members are much greater than between the two bias-correction methods. This is also summarised well by two last rows of each heat map panel for each metric: the standard deviation (summarised by the 'sd' row) across the ensemble is typically much larger than the difference between the BC methods in the 'mean' row. While over the reference period the BC significantly reduces the spread of the biases among the ensemble members (see row 'sd' in Figures 5 and 6), the spread in the projected changes is quite well retained by both BC methods (same row in Figures 11 and 12).

Although the differences in CCS between the two BC methods are relatively minor when compared to the differences between ensemble members for the same metric, there is a subtle added value of BCI3 over BCQM for preserving the changes, which is generally more visible looking at the individual ensemble members than in the ensemble means. It is worth noting that BCQM already largely preserves the projected changes, even though it was not explicitly designed to do so, which limits the potential benefit of even a perfect change preservation over this method. For temperature, the heatmaps in Fig. 12 show a limited added value of BCI3 compared to BCQM. This is in part due to the spatial averaging that took place in order to produce the heatmaps. Results for the climate change signal for each month and ensemble member for different quantiles (not shown) and Fig. 10 show that BCQM modifies the projected temperature climate change signal differently in different regions of the UK, in particular over significant topographical variation; this is not the case for BCI3, which better preserves the raw CCS spatial patterns. Finally, the differences in climate change signal between both BC methods are largest for several precipitation indices in winter, where BCI3 slightly improves the preservation of the projected change. Interestingly, for many of the temperature indices and the winter precipitation indices, the spread of projected changes is also better preserved in BCI3 than in BCQM (compare the ensemble 'sd' values). For potential evapotranspiration, similar analyses (figures not included here) show that BCI3 successfully reduced errors and conserved the climate change signal. Lastly, the heatmaps in Fig. 11 highlight the pitfalls of the change factor (CF) approach for UKCP18-RCM. There are large differences between the CF approach and the raw (or bias-corrected) projections when looking at the CCS in precipitation metrics beyond the seasonal or annual totals. The projected lengthening of the longest annual or summer dry day sequences and shortening of the longest annual, winter or summer wet day sequences in all ensemble members, which is well preserved by both BC methods, is largely or almost entirely disregarded using the CF approach. Most of the projected change in CDD and CWD can thus not simply be attributed to changes in the mean. The smaller projected decrease in the summer maximum 5-day accumulated precipitation (Rx5day) compared to the total precipitation decrease is not captured by the CF approach. Worse, at the annual scale the CF approach projects a decrease in the Q95 (similar to the prTOT changes), whereas the raw UKCP18-RCM projects an increase on average. At the annual scale, the CF approach is unable to capture the simulated change towards fewer (DF increase) but wetter (SDII increase) wet days, the combined effect of which is an overall decrease in total precipitation. This inability of the CF approach to account for changes in the variability of precipitation severely narrows its suitability for climate impact modelling studies on e.g. floods or droughts, for which the temporal variability and changes to the precipitation distribution are highly relevant.

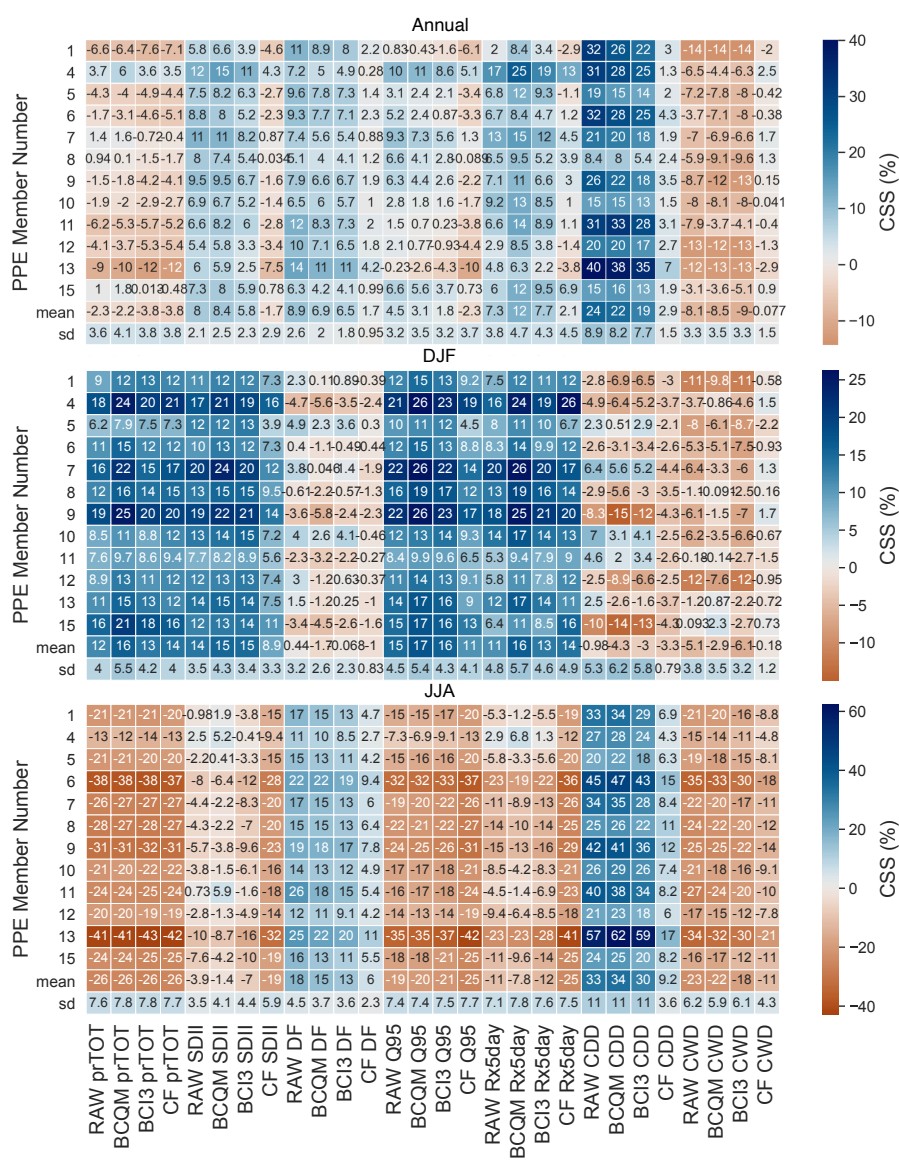

**Figure 11.** Projected changes (climate change signal; CCS) in precipitation characteristics in the ensemble of UKCP18-RCM simulations before (RAW) and after bias-correction (BCQM and BCI3) and after applying a change factor (CF) to the observed time series. Each indicator shows the spatially average (UK-mean) of the changes by 2051-2080 compared to 1981-2010, expressed as a percentage of the observed values for 1981-2010. Statistics shown are the same as in Fig. 5.



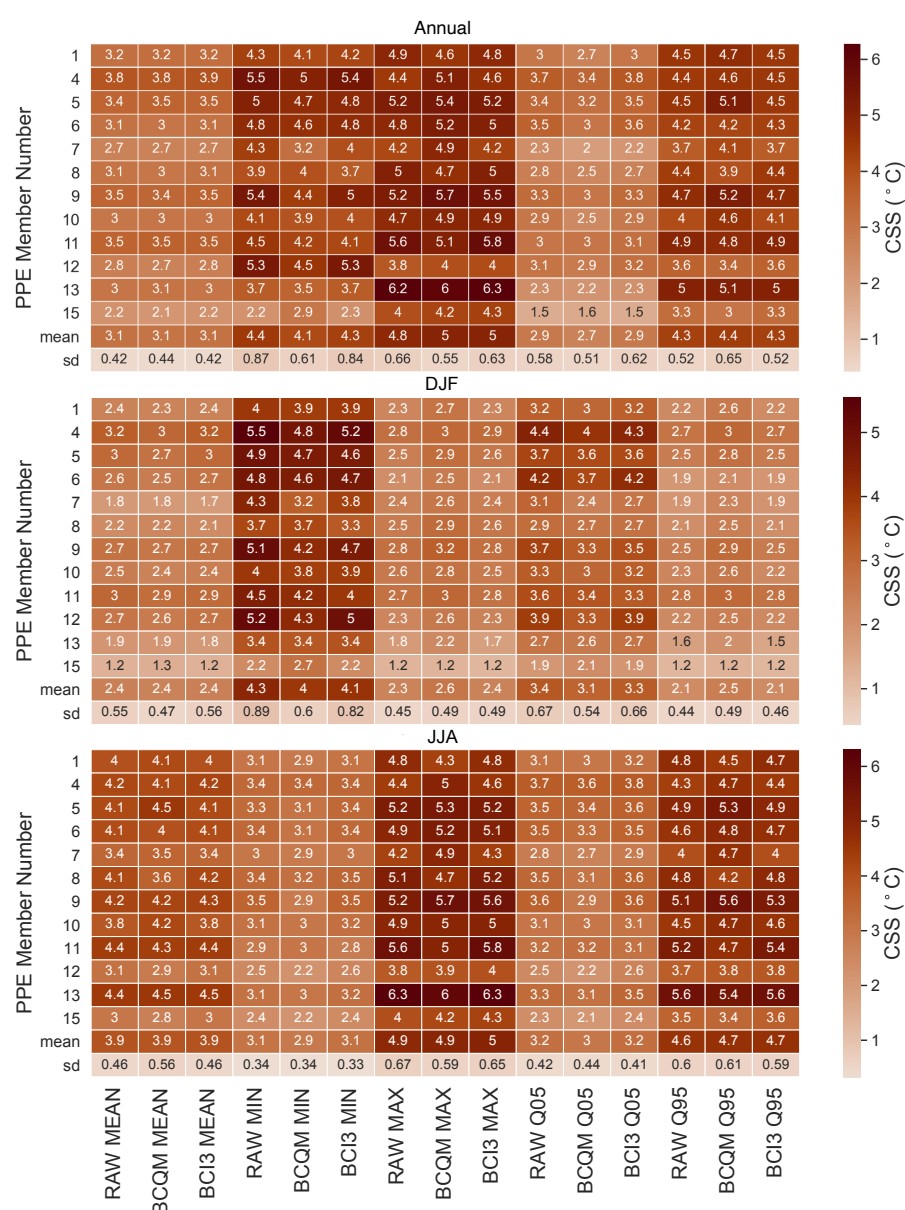

**Figure 12.** As Fig. 11 but for temperature characteristics and actual (°C) rather than percentage changes are shown. Statistics shown are the same as in Fig. 6.



## 4 Conclusions

### 4.1 General conclusions

Significant biases in temperature, potential evapotranspiration and precipitation statistics were found in the UKCP-RCM simulations. To improve their usability for impact modelling, two bias-correction methods were applied to all ensemble members of the UKCP-RCM perturbed physics ensemble: the widely used empirical quantile mapping (BCQM) and the bias-correction method developed for phase 3 of the ISIMIP project (BCI3), which is designed to preserve the climate change signal. Both methods successfully reduced errors in daily temperature and precipitation metrics, and reduced errors in multi-day precipitation metrics to a lesser degree. Both methods also satisfactorily conserved the climate change signal as well as the spread among the projections, with a minor improvement in BCI3 compared to BCQM. Analysis of projected changes in temperature and precipitation metrics suggests a higher likelihood of extreme weather (hot, dry or wet), and confirms the headline findings of projected hotter, drier summers and warmer, wetter winters.

### 4.2 Recommendations for users of the bias-corrected datasets

Potential users of these bias-corrected simulations are encouraged to consider the following points.

1. Both bias-corrected datasets may be of interest for users whose impact model is affected by the biases in the UKCP18-RCM simulations and whose application requires changes in the precipitation (and/or temperature or potential evapotranspiration) distribution and temporal variability to be captured as well as the mean. Conversely, using the change factor method is discouraged for applications where changes to the temporal variability and full distribution are important.

2. Caution is needed with coastal grid cells, as the 1 km data set used as reference for the bias-correction only covers the land, which results in biases in grid cell averages along the coast (especially for temperature and potential evapotranspiration). This should not be an issue for users only interested in land temperature or potential evapotranspiration, but we recommend that users consider the fraction of each grid cell covered by land before using the data.

3. At the regional scale, the climate change signal is slightly better preserved by BCI3 (especially for temperature). Although the differences between BCI3 and BCQM are small, they might grow or shrink after propagation in impact models, so it may be valuable to use both, as a limited way of sampling the uncertainty due to statistical postprocessing. However, the difference between BCQM and BCI3 is unlikely to be the greatest source of uncertainty.

4. PET time series are only provided for BCI3, so if using them alongside of temperature and/or precipitation time series, using BCI3 data only guarantees a consistent treatment of projected changes across variables.

5. Potential users of these bias-corrected datasets should also consult the UKCP18 user guidance published by the Met Office, in particular guidance on bias-correction (e.g. Fung et al., 2018). Since the uncertainty among the UCKP18-RCM ensemble members is large for changes in the precipitation and temperature metrics studied here, it is recommended to





employ multiple ensemble members to sample this uncertainty. Users are encouraged to perform their own evaluation of these datasets to ensure that they are adequate for their planned use.

6. As discussed in the introduction, all bias-correction methods rely on strong assumptions about climate projections, which users of any bias-corrected climate projections should keep in mind when using the data and interpreting the resulting impacts.




*Code and data availability.* This study produced bias-corrected UK Climate Projections 2018 (UKCP18) regional projections of temperature, precipitation, and potential evapotranspiration for 1981-2080. All 12 members of the UKCP18 regional ensemble were bias-corrected using the empirical quantile mapping method and a change-preserving variant of the ISIMIP3BA method. The access links for these datasets are provided below:

 – Daily temperature and precipitation data, bias-corrected by the empirical quantile mapping method, are available at https://zenodo.org/records/8223024 (Zha et al., 2023).

 – Daily temperature and precipitation data, bias-corrected using the ISIMIP3BA method are accessible at https://doi.org/10.5281/zenodo.6337381 (Reyniers et al., 2022a).

 – Bias-corrected daily potential evapotranspiration data can be found at https://zenodo.org/records/6320707 (Reyniers et al., 2022b).

The sources and links to the datasets used in this study:

 – CHESS-PE data was obtained from the UK CEH Environmental Information Data Centre (https://doi.org/10.5285/9116e565-2c0a-455b-9c68-558fdd9179ad) (Robinson et al., 2020b).

 – HadUK-Grid data was obtained from the Centre for Environmental Data Analysis (http://dx.doi.org/10.5285/d134335808894b2bb249e9f222e2eca8) (Met Office et al., 2019).

 – UKCP18-RCM simulations were acquired from the Centre for Environmental Data Analysis (https://catalogue.ceda.ac.uk/uuid/589211abeb844070a95d061c8cc7f604) (Met Office Hadley Centre, 2018).

The ISIMIP3BA code by Lange (2019) used in this study is accessible on Zenodo at https://zenodo.org/records/3898426 (Lange, 2020).

*Author contributions.* NR and QZ applied ISIMIP3BA and QM BC methods for bias-correction, respectively. NR, QZ, and NF performed the evaluations and assessment of projected changes. NR, NA, TJO and YH analysed results and drafted the manuscript. NF assisted with
data processing and figure producing. All authors reviewed the resulting inventory and assisted with paper writing.

*Competing interests.* The authors declare that they have no conflict of interest.

*Acknowledgements.* The authors are grateful for the support given by the high-performance Linux compute cluster at the University of East Anglia, which was used for conducting bias-correction in this study. We would like to express our gratitude to Lukas Gudmundsson, the developer of the QM R package 'qmap', for providing us with this valuable tool, as well as to Stefan Lange for his invaluable assistance with
the ISIMIP3BA BC method. The authors would also like to express their gratitude for the data made available by the Met Office (UKCP18, Had-UK Grid) and CEH (CHESS-PE).





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
