# Peer review of "Two sets of bias-corrected regional UK Climate Projections 2018 (UKCP18) of temperature, precipitation and potential evapotranspiration for Great Britain"

_Earth System Science Data, 2024_

## Author Response (AR1)

**Responses to all referees' comments**

We appreciate the feedback from the referees. We have carefully considered all comments to enhance the clarity, robustness, and scope of our study.

We also acknowledge the Editor's previous clarification that our manuscript falls within the journal's scope, despite concerns raised by the referees. We have ensured that our responses clearly demonstrate our adherence to the ESSD Data Description Paper guidelines and have focused on dataset quality, accessibility, and usability rather than introducing novel bias-correction methods. We provide a detailed response to the referees' comments.

Below, comments are marked in red.
Responses to the comments are marked in blue.
Manuscript changes are marked in *italic*.

Referee #1:

1. Line 35-40: This is a good and important discussion on key limitations of bias correction. However, the methodology used in this study did not addresses these challenges if your BC approach did help to "address the origin of model errors", that would be a significant contribution to the community. As it stands, the manuscript does not seem to offer solutions for tackling the root causes of model errors.

Thank you for raising this important point. We agree that addressing the origin of model errors would indeed represent a significant advance in the field. However, this is beyond the scope of our study, which focuses on providing a high-quality, bias-corrected dataset using established methods to support impact assessments, rather than addressing the underlying causes of these errors.

We refer to the manuscript types specified by ESSD (https://www.earth-system-science-data.net/about/manuscript_types.html). According to the journal's guidelines, data description papers in ESSD emphasize the quality, accessibility, and usability of datasets. Although demonstrating data quality is essential, extensive interpretations of data, as expected in a research article, fall outside the scope of this journal. ESSD specifically focuses on publishing high-quality research datasets to promote data reuse, rather than introducing novel methodologies or performing comprehensive comparisons.

Our study aligns with these goals by applying established bias-correction methods to improve the usability of climate projections for impact assessments. To further support the value of our work, we note that one of our datasets, available at https://zenodo.org/records/6337381, has already been downloaded 275 times as of 29/01/2025, demonstrating its significant contribution to the community.

2. Line 66: "simple BC methods used in CHESS-SCAPE": what methods they use?
We appreciate your request for clarification. In fact, an explanation is provided in Lines 60-65 of the manuscript. The CHESS-SCAPE dataset applies simple linear scaling, with additive

corrections for air temperature and multiplicative corrections for precipitation. Seasonal offsets are used for temperature, while scaling factors are applied for precipitation. Specifically, temperature offsets are calculated as the difference between CHESS-SCAPE and CHESS-met observations for each season and grid cell, and precipitation scaling factors are the ratio of CHESS-SCAPE to CHESS-met precipitation. We have revised the manuscript to clarify this method.

Lines 76-77 (manuscript with tracked changes): *However, the simple linear scaling BC method used in the CHESS-SCAPE does not take into account changes in higher-order moments than the mean.*

3. The data you corrected is from a perturbed physics ensemble, which is designed to both explore the influence of parameter variations on simulations and to reduce uncertainty. It would be highly valuable to examine how bias correction affects the ensemble results. For example, does using a single observation dataset to correct multiple ensemble runs impact the ensemble's representation of uncertainty? Additionally, it would be useful to see a discussion on the performance of individual ensemble members before and after bias correction. Such an analysis could provide insights into how bias correction interacts with model parameterizations and the ensemble spread, and provide physical interpretation of your results.

We appreciate this suggestion. In fact, we have already considered the effects of bias-correction on the ensemble spread, as seen in Fig. 4 showing the spread of monthly P/T before and after BC and figures for standard deviation ('sd' rows in Fig. 5, 6, 11, and 12) of reference period error and projected changes of a number of P/T indices. The individual ensemble members' performance and projected changes are also shown in these figures, showing how the bias-correction influences them. It is important to note that there are 12 sets of perturbed physics ensemble (PPE) parameters drawn from distributions of 47 parameters (as described in Section 2.1.1). This sparse coverage does not allow for deriving physical explanations of model behaviour or performance related to these perturbations, and the UKCP18-PPE-ensemble was not constructed with this purpose in mind.

4. The overestimation of dry-day frequency is a common issue in climate models, often linked to the drizzle effect. I recommend addressing the drizzle effect first, as this might improve the accuracy of subsequent bias corrections and separate this systematic bias from others.

Thank you for raising this point. Your comment might be referring to overestimation of wet-day frequency and hence the drizzle effect. The bias of dry-day frequency is already provided in Fig. 1 along with the text in Section 3.1.1 (Lines 231-240). Figure 1 shows the ensemble mean errors of the raw UKCP18-RCM projections in the dry-day frequency, mean daily precipitation and the Q95 of precipitation in the reference period, expressed as a percentage of the observed value. In general, the frequency of dry days in UKCP18-RCM is too low (and therefore the wet-day frequency is too high), particularly in the winter and in regions of higher elevation. In summer, the dry-day frequency bias is very small for most of England.

The precipitation mean and Q95 are strongly overestimated across the UK in winter, although in highly elevated areas this bias is smaller or even reversed in sign (especially for Q95). In summer, however, the mean and Q95 biases show a strong spatial variability, with underestimations toward the south and at high elevation levels, and a wet bias in the north of the UK. These seasonal bias differences result in an annual bias of too few dry days almost everywhere, too wet mean precipitation in most regions, and more mixed wet and dry Q95 relative biases.

The QM and trend-preserving BC methods employed in this study address the drizzle effect (overestimation of wet-day frequency) by matching the quantiles or re-distributing the precipitation values to align with the observed. In other words, both BC methods adjust the distribution of those lower-quantile values (light precipitation events) to align with the observed. The sections 2.4.1 and 2.4.2 already explained the methods and how the distribution is adjusted, therefore no further change is made in the manuscript.

Referee #2:

1. This paper evaluates two relatively simple statistical methods for correcting climate model output directly. The results indicate that bias correction techniques can mitigate biases related to the indices utilized in this study. Nonetheless, my principal concern is the paper's originality. Numerous studies have already been published that compare various statistical and machine learning methods across different domains and temporal contexts, ranging from historical data to future projections. Consequently, I do not believe this paper contributes significantly new information relative to existing literature, particularly given that available published data could offer additional insights. While the journal is dedicated to data publication, it is essential for this paper to either introduce a novel methodology or provide a comprehensive analysis of each method, highlighting the advantages and disadvantages of their application in specific circumstances. Merely comparing simple methods that produce comparable results for future projections may not provide substantial value.

Thank you for raising this important concern. We acknowledge that introducing a novel methodology would represent a significant contribution to the field. However, this is beyond the scope of our study, which focuses on providing a high-quality, bias-corrected dataset using established methods to support impact assessments, rather than introducing a new bias-correction method.

We refer to the manuscript types specified by ESSD (https://www.earth-system-science-data.net/about/manuscript_types.html). According to the journal's guidelines, data description papers in ESSD emphasize the quality, accessibility, and usability of datasets. Although demonstrating data quality is essential, extensive interpretations of data, as expected in a research article, fall outside the scope of this journal. ESSD specifically focuses on publishing high-quality research datasets to promote data reuse, rather than introducing novel methodologies or performing comprehensive comparisons. Our study aligns with these goals by applying established bias-correction methods to improve the usability of climate projections for impact assessments.

Finally, we would like to point out that even though the two bias-correction techniques are not novel, their application to the entire UKCP18 regional dataset (whole of UK, 12 members, 1981 to 2080, and three variables, including PET) is novel. The process of narrowing down

the selection of the two bias-correction methods, preprocessing the data, generating the bias-adjusted data, and evaluating the results for broader interests took about a year. This substantial effort ensures that research on climate change impacts in the UK can progress without duplicating efforts to bias-correct data that has already been carefully processed. The referee writes "Merely comparing simple methods that produce comparable results for future projections may not provide substantial value". We believe instead that these datasets do provide substantial value, and based on the fact that one of our datasets (https://zenodo.org/records/6337381) has been downloaded 275 times as of 29/01/2025, we would like to argue that the community sees this value too.

2. The manuscript lacks a comprehensive explanation for the preference of two simpler methods over more sophisticated approaches. The selection of the degree of bias correction should depend on the specific application, as advanced techniques have been shown to produce greater improvements, particularly in the context of extreme events. While quantile mapping and simple climatological mean correction have demonstrated their advantages in preventing excessive correction based on observational data, they may still permit the persistence of biases related to low-frequency variability, which can complicate the direct correction of surface variables in climate model outputs.

Numerous studies have employed sophisticated bias correction across various time scales to adjust the outputs of GCM and RCM or the boundary conditions for RCM inputs, with the objective of enhancing the accuracy of simulations for extreme and compound events. It would be advantageous to incorporate additional references that pertain to bias correction prior to addressing the limitations of existing studies. Furthermore, it is essential to provide detailed information and explanations regarding using simpler methods in the introduction or conclusion to ensure a thorough understanding of the techniques ranging from simple to sophisticated methods.

- Correcting outputs
Wood AW, Leung LR, Sridhar V, Lettenmaier D (2004) Hydrologic implications of dynamical and statistical approaches to downscaling climate model outputs. Clim Change 62:189–216
Cannon, A. J. (2018). Multivariate quantile mapping bias correction: an N-dimensional probability density function transform for climate model simulations of multiple variables. Climate dynamics, 50(1), 31-49.

- Correcting RCM input full variable fields
Bruyere CL, Done JM, Holland GJ, Fredrick S (2014) Bias corrections of global models for regional climate simulations of high-impact weather. Clim Dyn 43:1847–1856
Kim, Y, Evans, JP, Sharma, A (2023). Can Sub-Daily Multivariate Bias Correction of Regional Climate Model Boundary Conditions Improve Simulation of the Diurnal Precipitation Cycle?. Geophysical Research Letters, 50(22), p.e2023GL104442.

- Software for correcting climate model variables

Cannon, A.J. (2016). Multivariate bias correction of climate model output: Matching marginal distributions and intervariable dependence structure. Journal of Climate, 29(19), pp.7045-7064.

Kim, Y., Evans, J. P., & Sharma, A. (2023). A software for correcting systematic biases in RCM input boundary conditions. Environmental Modelling & Software, 168, 105799.

We agree that the selection of the bias-correction method should ideally be tailored to each specific impact study and the types of hydrological or meteorological extreme events being studied. However, barriers such as time/budget constraints, lack of knowledge of the available methods and the evolving field of bias adjustment, lack of validation/testing in impact modelling studies, and/or lack of available software implementing these innovative specialized bias-correction methods, can stand in the way of their uptake in many impact modelling studies. We make available two sets of bias-adjusted datasets for the community and provide results of a multi-metric evaluation, but as we emphasize in our conclusions section, in the end it is up to the user to decide which bias-correction method or bias-corrected dataset fits their specific purpose. We actively encourage potential users to perform their own evaluations in lines 431-432. (The significant added value of our work has been discussed in our response to major comment 1).

With the established quantile mapping method and the novel quantile mapping-based trend preserving bias-correction method developed for the 3rd phase of the ISIMIP-project, we believe we strike a good balance between simple and more sophisticated methods, as these (1) correct higher order moments (which commonly applied simpler methods don't do), and (2) have indeed proven themselves and have been put to the test in hydrological impact modelling studies.

Thank you for raising the concern of lower-frequency variability. To provide prospective users with insight on the degree to which multi-day metrics are corrected, we addressed this in the evaluation of the bias-adjusted datasets by looking at the errors and projected changes in the longest annual/seasonal streaks of consecutive wet days, consecutive dry days, and maximum 5-day total rainfall. As we discussed in L295-301, Fig. 5 shows that these are indeed more challenging to correct, but they are nevertheless improved to a great degree by both bias-adjustment methods.

We agree to discuss the suggested additional literature in the introduction and conclusions section.

Lines 32-44 (manuscript with tracked changes): *To this effect, a range of bias-correction (BC) methods have been developed and compared (Gutmann et al., 2014; Maraun et al., 2019; Teutschbein and Seibert, 2012; Wood et al., 2004). Our purpose here is not to provide an extensive review of alternative bias-correction methods, because such reviews are available elsewhere (see the introduction sections of Robertson et al. (2023) and Zhang et al. (2024) and the literature they cite for recent examples). These methods vary in complexity and scope, from univariate approaches to more advanced multivariate methods. For example, Cannon (2016) introduced a multivariate BC algorithm designed to correct inter-variable correlations, and later Cannon (2018) developed an n-dimensional multivariate quantile mapping BC method for a more comprehensive correction of multivariate dependence structures. Moreover, for completeness, we note that bias-correction is not only used as a processing step between climate model output and impact model, but is also sometimes*

*applied to correct the global climate model-derived boundary conditions used for dynamical downscaling with regional climate models (e.g. Bruyère et al., 2014). For example, Kim et al. (2023a) (software: Kim et al., 2023b) improved the simulation of diurnal precipitation cycles using their proposed sub-daily multivariate BC method. Generally speaking, bias adjustment methods transform the simulations so that some of their statistical properties match those of the observations.*

Lines 98-105 (manuscript with tracked changes): *In comparison to some more sophisticated proposed methods in literature (such as the examples discussed earlier in this introduction, which also rely on quantile mapping), the bias adjustment methods selected for the production of the datasets in this study are relatively straightforward, as they are univariate and correct only on the native daily time scale of the regional climate model simulations. However, these established quantile mapping-based methods provide substantial added benefits over the simplest bias adjustment methods and strike a good balance for the production of multi-purpose datasets from which the impact modelling community can benefit. The raw precipitation and temperature simulations and derived PET data were evaluated before the two BC methods were applied. The resulting bias-corrected datasets are also evaluated and compared, and finally, recommendations are made concerning the use of the datasets.*

Lines 456-459 (manuscript with tracked changes): *Users are encouraged to perform their own evaluation of these datasets to ensure that they are adequate for their planned use, for example if the correction of dependence structures of multiple variables is required (since both our methods are univariate and we did not evaluate multivariate metrics).*

3. The authors have undertaken corrections for three variables: precipitation, temperature, and potential evapotranspiration (PET). It would be beneficial to clarify the methodology employed for correcting PET. Did the authors correct PET derived from the raw variables directly, or did they utilize the adjusted variables in the PET calculations? As indicated by the authors, PET is influenced by several variables generated from the climate model, including specific humidity, pressure, and temperature, which can be modified by adjusting surface variables interconnected with these factors. Correcting these variables statistically, without accounting for the physical relationships among them, may lead to inconsistencies and produce unrealistic results.

Thank you for the comment. PET was calculated from the uncorrected (i.e. "raw") model variables (specific humidity, atmospheric pressure, net downwelling longwave and shortwave radiation, 10m wind speed and surface air temperature) and then the PET was bias-corrected against PET calculated from observed variables. One reason we did not do the alternative (i.e. bias correct the individual variables and then calculate PET from the bias-corrected individual variables) is that the PET might still show biases compared with observation-based PET because, as the referee notes, there may be inconsistencies between the individual variables in the model simulation. Taking this approach might then require a further bias-correction step to correct for remaining biases in the calculated PET, which our approach

avoids. We thank the referee for highlighting this question and we have added the following sentence to the end of section 2.2 (potential evapotranspiration) to make it clear to the reader:

Lines 150-154 (manuscript with tracked changes): *PET was calculated from the uncorrected model variables and then it was bias-corrected against PET calculated from observed variables (i.e. CHESS-PE). This was preferred to the alternative approach of bias correcting the individual variables and then calculating PET from the bias-corrected individual variables because that PET might still show biases compared with observation-based PET, requiring a further bias-correction step.*

Specific comments:
4. L66. "the simple methods …" It would be beneficial to provide more details about what these methods entail.
We appreciate your request for clarification. In fact, an explanation is provided in Lines 60-65 of the manuscript. The CHESS-SCAPE dataset applies simple linear scaling, with additive corrections for air temperature and multiplicative corrections for precipitation. Seasonal offsets are used for temperature, while scaling factors are applied for precipitation. Specifically, temperature offsets are calculated as the difference between CHESS-SCAPE and CHESS-met observations for each season and grid cell, and precipitation scaling factors are the ratio of CHESS-SCAPE to CHESS-met precipitation. We have revised the manuscript to clarify this method.

Lines 76-77 (manuscript with tracked changes): *However, the simple linear scaling BC method used in the CHESS-SCAPE does not take into account changes in higher-order moments than the mean.*

5. L68. "the quantile mapping (QM) method outperforms … the standards deviation and percentiles." Quantile mapping (QM) can outperform simple mean correction for variance since the latter does not address the standard deviation or percentiles. I recommend incorporating more details about bias correction techniques, including methods like simple mean and standard deviation correction. This will help justify the use of empirical QM when publishing datasets for broader applications.
Thank you for your comment. Several studies have conducted similar comparisons of bias-correction methods. We have revised the manuscript as follows:

*Lines 78-84 (manuscript with tracked changes): Several studies have compared different BC methods including linear scaling, delta change, local intensity scaling, variance scaling and QM. These studies consistently find that QM outperforms other BC methods in effectively correcting higher-order statistical properties, such as standard deviation and percentiles (Azmat et al., 2018; Teutschbein and Seibert, 2012; Worku et al., 2020). Specifically, Fang et al. (2015) and Enayati et al. (2021) highlighted the strengths of empirical QM for effectively correcting precipitation and temperature biases*

6. L82. "the trend-preserving BC method." It is essential to justify the trend-preserving method for future projections, as climate models also contain biases in trends.

We employed two methods. Both correct distribution biases: one does not explicitly consider trends in the projections, while the other one explicitly considers them and is designed to preserve them. Our results show that the projected changes are essentially insensitive to these two different approaches. This is illustrated by the future changes in the raw data and in the two bias-corrected datasets being overall similar (Fig. 9 and 10). As the referee mentions, the differences between these maps are "minimal". Importantly, they are much smaller than differences between members (see e.g. Fig. 12). In other words, larger uncertainties are introduced by running an ensemble than by treating future trends differently. In addition, the Met Office performed an evaluation of long-term drift in the UKCP18 global and regional PPE simulations. Optionally, we can also discuss this report in the "data" section to address this comment.

7. Figures 9 and 10. I recommend modifying the figures to use a bias map instead of presenting each one individually, as the differences are minimal.

Thank you for your comment. While we recognize that the differences between the raw and bias-corrected data in Fig. 9 and Fig. 10 are small, these figures serve two purposes, which are more efficiently fulfilled by the current representation. First, they demonstrate that the projected changes are retained after bias-correction. Second, they are crucial for illustrating the spatial distribution of temperature and precipitation changes within the UKCP18 model. This spatial representation provides valuable insights into how these projected changes across different regions, and the minor distortions of the spatial coherence of projections by the quantile mapping method in the context of the magnitude of the projected changes (e.g. for summer temperature in Fig. 10), which would not be as clear in a bias map. Therefore, we believe that maintaining the current figure more effectively communicates both the preservation of projected changes and their regional variability.

In addition to the revisions made in response to the referees' comments, we have implemented the following minor modifications:

1. The affiliation "School of Environmental Sciences, University of East Anglia, Norwich, UK" has been removed.
2. In lines 45-46 (manuscript with tracked changes), 'after the correction' has been modified to '*after bias correction*' for improved clarity.
3. In line 63 (manuscript with tracked changes), 'Gudde et al., in review' has been updated to '*Gudde et al., 2024*'.
4. In line 87 (manuscript with tracked changes), the article 'a' has been added before 'snow module'.
5. In line 111 (manuscript with tracked changes), 'section 3.2' has been replaced with '*Section 3.2*'.
6. In line 461 (manuscript with tracked changes), 'users' has been modified to '*all users*'.

**References**

[revised manuscript text omitted]